# Finding the Common Single-Nucleotide Polymorphisms in Three Autoimmune Diseases and Exploring Their Bio-Function by Using a Reporter Assay

**DOI:** 10.3390/biomedicines11092426

**Published:** 2023-08-30

**Authors:** Yen-Chang Chu, Kuang-Hui Yu, Wei-Tzu Lin, Wei-Ting Wang, Ding-Ping Chen

**Affiliations:** 1Department of Ophthalmology, Chang Gung Memorial Hospital at Linkou, Taoyuan 333, Taiwan; yenchang@adm.cgmh.org.tw; 2College of Medicine, Chang Gung University, Taoyuan 333, Taiwan; goutyu@gmail.com; 3Division of Rheumatology, Allergy, and Immunology, Chang Gung Memorial Hospital at Linkou, Taoyuan 333, Taiwan; 4Department of Laboratory Medicine, Chang Gung Memorial Hospital at Linkou, Taoyuan 333, Taiwan; berry0908@cgmh.org.tw (W.-T.L.); s1223@cgmh.org.tw (W.-T.W.); 5Department of Medical Biotechnology and Laboratory Science, College of Medicine, Chang Gung University, Taoyuan 333, Taiwan

**Keywords:** systemic lupus erythematosus, rheumatoid arthritis, Graves’ ophthalmopathy, single-nucleotide polymorphisms, common, bio-function

## Abstract

In clinical practice, it is found that autoimmune thyroid disease often additionally occurs with systemic lupus erythematosus (SLE) and rheumatoid arthritis (RA). In addition, several studies showed that eye-specific autoimmune diseases may have a strong relationship with systemic autoimmune diseases. We focused on Graves’ disease (GD) with ocular conditions, also known as Graves’ ophthalmopathy (GO), trying to find out the potential genetic background related to GO, RA, and SLE. There were 40 GO cases and 40 healthy controls enrolled in this study. The association between single-nucleotide polymorphisms (SNPs) of the co-stimulatory molecule genes and GO was analyzed using a chi-square test. It showed that rs11571315, rs733618, rs4553808, rs11571316, rs16840252, and rs11571319 of *CTLA4*, rs3181098 of *CD28*, rs36084323 and rs10204525 of *PDCD1*, and rs11889352 and rs4675379 of *ICOS* were significantly associated with GO based on genotype analysis and/or allele analysis (*p* < 0.05). After summarizing the GO data and the previously published SLE and RA data, it was found that rs11571315, rs733618, rs4553808, rs16840252, rs11571319, and rs36084323 were shared in these three diseases. Furthermore, the bio-function was confirmed by dual-luciferase reporter assay. It was shown that rs733618 T > C and rs4553808 A > G significantly decreased the transcriptional activity (both *p* < 0.001). This study is the first to confirm that these three diseases share genetically predisposing factors, and our results support the proposal that rs733618 T > C and rs4553808 A > G have bio-functional effects on the transcriptional activity of the *CTLA4* gene.

## 1. Introduction

Systemic lupus erythematosus (SLE), rheumatoid arthritis (RA), and Graves’ disease (GD) have many common characteristics, including their prevalence in women and the production of autoantibodies due to the over-activation of autoreactive T cells and the over-proliferation of B cells [1]. Additionally, it was found that thyroid dysfunction is common in SLE and RA [2,3]. Many patients begin treatment for thyroid dysfunction before they are diagnosed with lupus or RA, and vice versa [4]. Furthermore, studies found that there was serological overlap among SLE, RA, and autoimmune thyroid disease (AITD), such as thyroid autoantibodies (ThyAb), thyroid-stimulating hormone (TSH), triiodothyronine (T3), thyroxine (T4), free triiodothyronine (fT3), free thyroxine (fT4), and so on [2,3,5]. Moreover, it was found that treating GD with methoxazole or propylthiouracil could induce the development of SLE [1]. Furthermore, hydroxychloroquine is a commonly used medication for RA and SLE to help control the symptoms [6]. These findings indicated a possible common mechanism among SLE, RA, and GD.

In recent years, more and more studies have shown that SLE is associated with AITD, including GD and Hashimoto’s thyroiditis [7]. In addition, a recent study indicated that GD was associated with an increased risk of SLE, which suggested that there may be an inseparable relationship between AITD and lupus disorders [8]. In a prospective study in 1987, abnormal thyroid function was found frequently in SLE patients [9], which indicates that the association between AITD and SLE has been reported for more than 50 years. Furthermore, Wu et al. showed that the pathogeneses of RA and GD were interrelated [10]. Although AITD has been reported individually with SLE and RA for many years, the common pathogenesis is still not well understood.

It is clinically shown that about one-third of SLE patients will have ocular complications [11]. Eye symptoms may relate to systemic disease activity and can be used as an initial manifestation of SLE [12]. There are two major types of AITD: Graves’ disease and Hashimoto’s autoimmune thyroiditis. Eye involvement in GD has been named Graves’ ophthalmopathy (GO). GO is characterized by swelling of the orbital tissue in GD patients. A genetic factor is believed to be a risk factor for GO. According to statistics, 50% of GO patients have a family history [13]. Furthermore, compared with GD patients without ocular symptoms, GO patients had a higher frequency of catching other autoimmune diseases [14]. This suggests that eye-specific autoimmune diseases may have a stronger relationship with systemic autoimmune diseases than other AITDs. Therefore, we focused on the association between GO and other autoimmune diseases in this study. When we set out, there was no study on the correlation between GO, RA, and SLE, so we sought to determine the potential pathogenesis related to these three diseases.

SLE is a systemic autoimmune disease, which is mainly caused by the loss of immune tolerance and immune imbalance led by genetic factors [15]. GD is also an autoimmune disease, which is caused by the excessive secretion of thyroid hormone due to the production of thyrotropin receptor antibody (TRAb), and a genetic factor is one of the risk factors for the pathogenesis of GD [16]. In addition, RA is also an autoimmune disease associated with genetic susceptibility. The heritability of RA is up to 50–60%, which indicates that a genetic factor plays a vital role in the pathogenesis of RA [17]. Although the pathogenesis of SLE, RA, and GO is still unclear, genetic factors are considered to be the key query point. We previously studied the association between SNPs of the co-stimulation molecule genes and SLE [18] and RA [19]. In this study, we determine the common SNPs in these three autoimmune diseases by consolidating the SNP analysis data of SLE, RA, and GO and further verify the biological function of the SNP with statistical significance.

## 2. Materials and Methods

### 2.1. Inclusion Criteria

The diagnosis of GO is made when 2 of the following 3 signs of the disease are present: (1) Circulating thyroid antibodies or a dysthyroid state. (2) Typical ocular signs. (3) Fusiform enlargement of extraocular muscles. The inclusion criteria of the healthy control group were those without autoimmune diseases, immune abnormalities, or using immunosuppressive drugs. A total of 100 volunteers were recruited as control cases in the same IRB, and the same number of control cases was taken from those for SNP analysis.

### 2.2. Selection of Candidate SNPs

Because these autoimmune diseases are caused by abnormal immune regulation, we explored the SNPs of the co-stimulatory molecule genes, which are involved in the regulation of T-cell activation, including *CTLA4*, *CD28*, *PDCD1*, *TNFSF4*, and *ICOS*. Previously, only the CTLA4 gene polymorphism and its correlation were analyzed in GO patients [20]. In this study, the GO sample size was increased, and we took the candidate SNPs in the previously published association study between SLE/RA and the genetic polymorphisms of the co-stimulatory system [18,19] as the candidate SNPs of GO, to explore the association between these SNPs and GO. Please refer to ref. [18,19] for the primers and PCR programs used.

### 2.3. DNA Extraction and Sequencing

The genomic DNA was extracted from 200 µL of peripheral blood using a QIAamp DNA Blood Mini Kit (Qiagen, Hilden, Germany). Then, the concentration and purity of the extracted DNA were measured using a UV spectrometer before polymerase chain reaction (PCR). PCR was carried out in a total volume of 25 μL containing 50 ng of DNA, 7.5 µL of Hotstar Taq DNA Polymerase (Qiagen, Hilden, Germany) or 2X Tag polymerase, 1 µL each of forward and reverse primer (10 μΜ), and 14.5 µL of ddH_2_O. The primer pairs of each gene region and the PCR programs were the same as in the previous study [18,19]. After verifying the DNA fragments produced by PCR through gel electrophoresis, the Big Dye Terminator Cycle Sequencing kit (Thermo Fisher, Waltham, MA, USA) and the ABI PRISM genetic analyzer (Thermo Fisher, Waltham, MA, USA) were used for direct sequencing according to the manufacturer’s instructions.

### 2.4. Promoter–Reporter Construction

First, we found a sample from the included cases with the C_rs11571315_T_rs733618_A_rs4553808_C_rs16840252_ haplotype, which we used as the wild type. The promoter region of the CTLA4 gene in this sample was amplified by using the primer with HindIII and SacI restriction enzyme cleavage sites. The sequence of promoter fragments was confirmed by using ABI PRISM 3730 DNA analyzer (Applied Biosystems, Foster City, CA, USA). Then, the fragments were transferred into competent cells (Top 10 or DH5α) through the TOPO TA Cloning Kit (Invitrogen, Carlsbad, CA, USA). After culturing the competent cells, the plasmid DNA was extracted by X-gal, and the sequence of plasmid DNA was checked using direct sequencing. This plasmid DNA was used as the template for creating a single SNP variation via site-directed mutagenesis PCR (Quick Change Site-Directed Mutagenesis Kit, Stratagene, La Jolla, CA, USA). The pairs of primers are shown in Table 1.

### 2.5. Cell Culture and Transient Transfections

We routinely cultured 1 × 10^6^ K562 cells in 90% RPMI 1640 medium supplemented with 10% fetal bovine serum, penicillin (50 U/mL), and streptomycin (50 μg/mL) for follow-up experiments. The promoter–reporter constructs were transferred to the pNL1.1 [Nluc] expression vector (Promega, Madison, WI, USA) with NanoLuc luciferase. Similarly, these vectors were transferred into competent cells and confirmed by direct sequencing. Next, 1 μg of the pNL1.1 NanoLuc expression vector with the wild-type sequence or single SNP variation and 1 μg of the pGL 4.5 firefly expression vector (Promega) were transfected into 400 μL (2.5 × 10^5^) K562 cells together by using Lipofectamine 2000 (Invitrogen) and cultured, with pGL 4.5 used as the internal control to exclude bias in the transfection efficiency.

### 2.6. Dual-Luciferase Reporter Assay

After culturing for 24 h, these cells were detected using a Luciferase Assay System (Nano-Glo^®^ Dual-Luciferase^®^ Reporter Assay System, Promega) according to the manufacturer’s protocol. Each promoter–reporter assay was conducted 5–6 times in parallel. The luminescence of NanoLuc luciferase was divided by Firefly luciferase to exclude bias. In addition, the value of the wild type was referenced as 1 to compare the relative light units (RLUs) of each SNP variation.

### 2.7. Statistical Analysis

Before all analyses, the genotype contributions of all genes in the control group were analyzed using the Hardy–Weinberg equilibrium (HWE) to confirm that the included control group was representative of the entire population. Then, the allele and genotype contributions were analyzed using the chi-square test or Fisher’s exact test when the expected value of more than 20% of the cells was less than 5, given the odds ratio (OR) with a 95% confidence interval (CI). Among them, the lower-frequency allele was known as the minor allele, which was used to assess the effect of people with a minor allele on disease development. For multiple comparisons, the false discovery rate (FDR) Q-values were calculated to evaluate the expected proportion of type I errors. The haploid blocks were identified by linkage disequilibrium (LD) analysis, which was defined according to the definition proposed previously by Gabriel et al. [21]. We deleted the haplotypes with frequencies of less than 0.01. ANOVA and Tukey’s honestly significant difference test were used to analyze the difference between the RLU of the wild type and the vector with a single SNP variation. The statistically significant differences were considered as *p* < 0.05.

## 3. Results

### 3.1. Study Subjects

The GO cases (45.5 ± 15.2 years old) comprised 18 males (45%) and 22 females (55%), totaling 40 cases. The control group (37.6 ± 6.8 years old) for GO contained 7 males (18%) and 33 females (72%).

### 3.2. Hardy–Weinberg Equilibrium Test

First, the genotype frequencies of every SNP from the control group were analyzed using the Hardy–Weinberg equilibrium (HWE) to eliminate statistical errors. It was found that most SNPs satisfied the HWE; only rs3181096 of *CD28* and rs10932035 and rs11571305 of *ICOS* deviated from the HWE (Table 2). Therefore, these three SNPs were excluded from the subsequent SNP analysis and discussion.

### 3.3. Allele and Genotype Analysis

The allele frequencies are shown in Table 2. The GO-associated SNPs are shown in Table 3 and the complete data are shown in Appendix A. In addition, the Q-values were calculated to evaluate the proportion of significant tests that will result in false positives (Appendix A). In the *CTLA4* gene, six SNPs had statistical significance: rs11571315, rs733618, rs4553808, rs11571316, rs16840252, and rs11571319. The genotypes of rs11571315 were significantly different between GO cases and healthy controls (CC vs. CT vs. TT, *p* = 0.006, Q = 0.0720). Compared to TT, the CT genotype had 0.327 times lower odds (95% CI = 0.123–0.870, *p* = 0.023, Q = 0.1712) and it had 0.077 times lower the odds of CC genotype exposure (95% CI = 0.009–0.682, *p* = 0.015, Q = 0.1675). In addition, when people had at least one C-allele (CT + CC), they had lower odds of GO (OR = 0.257, 95% CI = 0.101–0.652, *p* = 0.004, Q = 0.0766). In allele analysis, the C-allele of rs11571315 had 0.291 times lower odds of GO (95% CI = 0.138–0.612, *p* = 0.001). In other words, the T-allele in rs11571315 was a risk allele of GO. The genotypes of rs733618 had significant differences between GO cases and controls (CC vs. CT vs. TT, *p* = 0.011, Q = 0.0977). Compared to CC, the TT genotype (OR = 0.135, 95% CI = 0.034–0.545, *p* = 0.003) and genotype with at least one T-allele (CT + TT, OR = 0.297, 95% CI = 0.094–0.934, *p* = 0.0032) had lower odds of GO. Moreover, people with TT in rs733618 had lower odds of GO than those with CT and CC (OR = 0.239, 95% CI = 0.082–0.696, *p* = 0.007, Q = 0.1173). In allele analysis, the T-allele of rs733618 had lower odds of GO (OR = 0.378, 95% CI = 0.199–0.717, *p* = 0.003). The genotypes of rs4553808 had statistical significance (AA vs. AG, *p* = 0.019, Q = 0.0977). In this SNP, there were only two genotypes, AA and AG, found in the population included in this study. Compared to AA, people with AG had 0.214 times lower odds (OR = 0.214, 95% CI = 0.055–0.838, *p* = 0.019, Q = 0.1675). In allele analysis, the G-allele of rs4553808 had lower odds of GO (OR = 0.244, 95% CI = 0.065–0.912, *p* = 0.025). The genotypes of rs11571316 were near statistical significance (GG vs. AG vs. AA, *p* = 0.056, Q = 0.1833). Compared to GG, people with the AG genotype had 0.321 times lower odds of GO (95% CI = 0.112–0.916, *p* = 0.030, Q = 0.1787) and people with at least one A-allele (AG + AA) had 0.306 times lower odds of GO (95% CI = 0.113–0.826, *p* = 0.017, Q = 0.1675). In allele analysis, the A-allele of rs11571316 had lower odds of GO (OR = 0.356, 95% CI = 0.152–0.836, *p* = 0.015). The genotypes of rs16840252 were significantly different between GO cases and controls (CC vs. CT, *p* = 0.019, Q = 0.0977). In this SNP, there were only two genotypes, CC and CT, found in the population included in this study. Compared to CC, the CT genotype had lower odds of GO (OR = 0.214, 95% CI = 0.055–0.838, *p* = 0.019, Q = 0.0977). In allele analysis, the T-allele of rs16840252 had lower odds of GO (OR = 0.244, 95% CI = 0.065–0.912, *p* = 0.025). The genotypes of rs11571319 located in 3′UTR of CTLA4 were significantly different between cases and controls (GG vs. AG vs. AA, *p* < 0.001, Q = 0.0178). Compared to GG, people with the AG genotype (OR = 0.123, 95% CI = 0.042–0.360, *p* < 0.001, Q = 0.0302) or at least one A-allele (OR = 0.118, 95% CI = 0.040–0.344, *p* < 0.001, Q = 0.0302) had lower odds of GO. In allele analysis, the A-allele of rs11571319 had lower odds of GO (OR = 0.178, 95% CI = 0.069–0.464, *p* < 0.001).

Regarding the *CD28* gene, there were two SNPs associated with GO, rs3181097 and rs3181098. The genotypes of rs3181097 were significantly different between cases and controls (GG vs. AG vs. AA, *p* < 0.001, Q = 0.0178). Compared to AA, people with the GG genotype (OR = 0.027, 95% CI = 0.003–0.249, *p* < 0.001, Q = 0.0302) or at least one G-allele (OR = 0.041, 95% CI = 0.005–0.331, *p* < 0.001, Q = 0.0302) had lower odds of GO. In allele analysis, the G-allele of rs3181097 had lower odds of GO (OR = 0.301, 95% CI = 0.157–0.578, *p* < 0.001). The genotypes of rs3181098 were close to being statistically significant (GG vs. AG vs. AA, *p* = 0.055). Moreover, the genotype frequencies of AG + GG and AA were significantly different between GO cases and controls (GG + AG vs. AA, *p* = 0.026, Q = 0.1742). Because there were no GO cases with the AA genotype, the odds ratio is not shown.

Regarding the *PDCD1* gene, two SNPs had statistical significance, rs36084323 and rs10204525. Compared to the TT genotype, people with the CC genotype in rs36084323 had higher odds of GO (OR = 4.125, 95% CI = 1.057–16.097, *p* = 0.037, Q = 0.1983). Compared to the TT + CT genotype, people with the CC genotype in rs10204525 had higher odds of GO (OR = 2.688, 95% CI = 1.076–6.715, *p* = 0.032, Q = 0.1787). In allele analysis, the C-allele of rs36084323 and rs10204525 had a higher risk of catching GO (OR = 2.048, 95% CI = 1.061–3.955, *p* = 0.032 and OR = 2.427, 95% CI = 1.172–5.023, *p* = 0.015, respectively).

Regarding the ICOS gene, two SNPs had statistical significance, rs11889352 and rs4675379. The genotypes of rs11889352 were significantly different between cases and controls (AA vs. AT vs. TT, *p* = 0.045, Q = 0.1800). No matter whether comparing to the AA genotype or AG + AA, people with TT had higher odds of GO (AA vs. TT, OR = 10.733, 95% CI = 1.197–96.283, *p* = 0.020, Q = 0.1675; AA + AT vs. TT, OR = 8.531, 95% CI = 0.997–73.006, *p* = 0.029, Q = 0.1787). In allele analysis, the T-allele of rs11889352 had higher odds of GO (OR = 2.272, 95% CI = 1.135–4.546, *p* = 0.019). The genotypes of rs4675379 were significantly different between cases and controls (GG vs. CG vs. CC, *p* = 0.042, Q = 0.1800). Compared to the GG genotype, people with the CG (OR = 0.278, 95% CI = 0.090–0.859, *p* = 0.023, Q = 0.1712) or at least one C-allele (OR = 0.333, 95% CI = 0.111–1.001, *p* = 0.047) had lower odds of GO. However, the allele contributions of rs4675379 were not significantly different between cases and controls (*p* = 0.178).

### 3.4. Haplotype Analysis

In Figure 1, the color of the box indicates the degree of linkage disequilibrium (LD) between the two SNPs. The color gradually changes from white to red, indicating that LD is becoming stronger, and purple indicates that there is no LD. In LD analysis, it was found that *CD28* (rs1879877/rs3181097/rs3181098), *CTLA4* (rs62182595/rs16840252/rs5742909), and *PDCD1* (rs2227981/rs2227982/rs6705653/rs41386349) each had one haplotype block, and *ICOS* had two haplotype blocks (rs11889352/rs11883722 and rs10932036/rs4404254/rs10932037/rs10932038). In haplotype analysis (Table 4), it was found that the five CTLA4 haplotypes (A_rs62182595_T_rs16840252_C_rs5742909_, A_rs62182595_T_rs16840252_T_rs5742909_, A_rs62182595_C_rs16840252_C_rs5742909_, A_rs62182595_C_rs16840252_T_rs5742909_, and G_rs62182595_T_rs16840252_C_rs5742909_) and one ICOS haplotype (A_rs11889352_C_rs11883722_) were associated with GO (all *p* = 0.034).

### 3.5. Transcriptional Activity Analysis

After integrating the data of SLE, RA, and GO, it was found that rs11571315, rs733618, rs4553808, rs16840252, and rs11571319 of *CTLA4* and rs36084323 of *PDCD1* had a significant statistical association in these three autoimmune diseases (Table 5). Then, the dual-luciferase reporter assay was used to explore the influence of SNP variation in the promoter region of the *CTLA4* gene on transcriptional activity.

The bio-function of the significant SNPs located in the *CTLA4* promoter region was analyzed through dual-luciferase reporter assay. It was shown that rs733618 T > C and rs4553808 A > G had a significant effect on transcriptional activity, but rs11571315 C > T and rs16840252 C > T did not (Table 6 and Figure 2). The C-allele of rs733618 had 0.263 times lower transcriptional activity than the T-allele (*p* < 0.001), and the G-allele of rs4553808 reduced the transcriptional activity level to 0.245 times that of the A-allele (*p* < 0.001).

## 4. Discussion

Previously, we found that rs733618 of *CTLA4* was significantly associated with GO and rs16840252 had a strong tendency towards statistical significance based on the data from 22 GO cases and 20 healthy controls [20]. In this study, the sample size was increased to 40 GO cases and 40 healthy controls. In addition, the data about SLE and RA that were previously published [18,19] and the data on GO in this study were combined to find out the common SNPs among these three diseases.

In 2019, we found that rs733618 of *CTLA4* was significantly associated with GO based on data from 22 GO cases and 20 healthy controls, while rs16840252 had a strong tendency towards statistical significance [20]. Here, we increased the sample size to 40 GO cases and 40 healthy controls, and it was found that rs11571315, rs4553808, and rs11571319 of the *CTLA4* gene were also associated with GO in addition to rs733618 and rs16840252. Most studies found that rs231775 of CTLA4 was associated with GO [22], but our study did not. A meta-analysis showed that rs231775 was associated with GO, which was more significant in European populations than in Asian populations [22]. Thus, there may be differences between ethnic groups. In addition, other significant SNPs had only been reported related to GD rather than GO. For example, rs733618 was found to be associated with GD in the Taiwanese population [23]; rs11571315 was found to be associated with GD in the Chinese Han population [24]; and rs11571319 was associated with GD when combined with other SNPs into a haplotype [25]. It could be seen that *CTLA4* was undoubtedly one of the susceptibility genes for GD or further development into GO; however, its variants associated with GD/GO varied widely across populations. Concerning our research about the correlation between GD and *CTLA4* polymorphism, it was found that rs733618 T/C and rs231775 G/A were associated with GD [20]. In addition to rs733618, we also found that rs11571319 was associated with GO. Thus, rs11571319 may be a susceptibility SNP specific to GO, rather than GD.

According to the available information, there was no literature about the association between the SNPs of rs4553808, rs16840252, rs36084323, rs10204525, rs3181098, rs11889352, and rs4675379, and GD/GO. Although there was no literature associated with GO or GD, these SNPs were associated with other autoimmune diseases or cancers. It was found that rs4553808 was significantly correlated with Hashimoto’s thyroiditis disease, which is also a thyroid disease [26]; rs16840252 was related to the risk of colon cancer [27]; rs10204525 was related to Posner–Schlossman syndrome, an orbital disease, when it was integrated with other SNPs [28]; rs3181098 was associated with malignant melanoma and its metastasis-free survival rate reduction [29]; and rs4675379 was associated with coeliac disease [30]. It shows that these SNPs also have specific functions in immune regulation. However, rs11889352 has no relevant research at present. Meanwhile, it is known that the promoter activity of rs36084323 with the A-allele is lower than the G-allele, and it may cause various autoimmune thyroid diseases by affecting the expression of PD-1 on Treg cells, the expression of PD-1/PD-1 ligand (PD-L1) on thyroid, and the titers of thyroglobulin autoantibody [31]. Therefore, rs36084323 may be an important hub of thyroid disease.

After integrating the data of SLE, RA, and GO, it was found that several SNPs had intersections, including rs11571315, rs733618, rs4553808, rs16840252, and rs11571319 of *CTLA4* and rs36084323 of *PDCD1*, which indicated that these three diseases had a partial genetic background. Thus, these SNPs may play an important role both in the pathogenesis of systemic autoimmune diseases (such as SLE and RA) and eye-specific autoimmune diseases (such as GO). Since they share many features, it was not surprising that they shared the same genetic predisposing factors. CTLA4 and PDCD1 are important negative regulators of T-cell activation [32]. As mentioned in the first paragraph of Section 4, these SNPs may also be susceptible to other autoimmune diseases and cancers. It shows that negative regulation of T cells may be more important than positive regulation in the pathogenic mechanism of autoimmune diseases. The haplotypes with statistical significance of SLE, RA, and GO contained rs62182595 and rs16840252 of *CTLA4*, leading us to surmise that these two SNPs may have an interaction with the key SNP that causes the disease. They were significant in SNP analysis, but it was not real pathogenic SNPs. This conjecture was verified in our functional analysis. The SNP variation of rs16840252 did not affect the transcriptional activity of the CTLA4 gene. In the functional analysis, it was found that rs733618 T > C and rs4553808 A > G significantly reduced the transcriptional activity. In addition, the transcriptional activity analysis of rs36084323 of *PDCD1* was conducted in our previous study [33], and it was found that rs36084323 C > T would decrease the transcription activity by 0.68 ± 0.07 times. In the SNP analysis, it was found that rs733618 T-allele and rs4553808 G-allele had a lower risk of SLE, RA, and GO. Theoretically, the decreased expression of CTLA4 contributes to autoimmune disease. Therefore, the higher gene expression level may explain the association of rs733618 T-allele with a lower risk of various autoimmune diseases. Our results proved that the rs733618 C-allele had lower transcriptional activity, which was the same finding as that of our research team [34,35,36]. Moreover, an eQTL analysis by Cai et al. showed that rs733618 could function as a cis-eQTL to affect membrane CTLA4 or total CTLA4 expression in the hippocampus [37], and cis-eQTL can affect the majority of human genes rather than specific tissue [38]. Thus, rs733618 seems to be a key SNP regulating CTLA4 expression level. In this study, it was found that the rs4553808 G-allele decreased the transcriptional activity of CTLA4, and Kaykhaei et al. also demonstrated that rs4553808 in the presence of the G-allele was the transcription factor binding sites of CCAAT-enhancer-binding protein β and glucocorticoid receptor [26], thereby up- or down-regulating the transcription of CTLA4. However, the rs4553808 G-allele decreased the risk of SLE, RA, and GO, which was rather illogical. After integrating these results, we found that more than one SNP in a gene could regulate the gene transcriptional level at the same time, and we inferred that the final protein expression level should be the integration of these functional SNPs. Therefore, it was not enough to demonstrate that SNP affected the occurrence of diseases only by looking at the effect of specific sites on gene expression. It was found from our results that the allele changes in rs11571315 and rs16840252 would not affect their transcriptional activity. The single-tissue expression quantitative trait loci (eQTL) analysis showed that the allele variation of rs11571315 only influenced the expression level of CTLA4 in certain tissues, such as the esophagus, testis, heart, and artery [39], which indicated that the gene expression changes caused by rs11571315 may be tissue-specific. At present, it has not been suggested that rs16840252 is functional, and it often had a strong LD with other susceptibility SNPs or was associated with disease susceptibility after being combined into a haplotype [18,20,28,40,41,42]. Therefore, it was speculated that rs16840252 was statistically significant in SNP analysis because of its strong linkage imbalance with susceptibility SNPs, or it will be functional after interacting with other SNPs. In addition, the mechanism of other diseases related to the meaningful SNPs found in functional analysis may also be due to their regulation of gene transcription activity.

In the future, large-scale and carefully designed research should be carried out, taking into account detailed environmental factors, to confirm this relationship in different populations, so as to further verify these associations, especially for gene–environment and gene–gene interactions, or researchers could select T cells with specific SNPs or haplotypes from patients to culture in vitro to test the CTLA4-mediated immunosuppression. In addition, functional analysis of the promoter SNP could verify that these SNPs affected the transcriptional function of the gene and were associated with the occurrence of the disease. rs733618 and rs4553808 were related and had a biological function in three autoimmune diseases at the same time, indicating that these two SNPs may play an important role in the mechanism of these autoimmune diseases, which could provide a new direction for their treatment. However, the allele frequencies of these common SNPs of *CTLA4* had no statistical significance in RA but were only associated with RA in the heterozygous genotype [19], which could indicate that the pathogenesis of RA caused by *CTLA4* SNPs may be different from that of SLE and GO. Moreover, the human leukocyte antigen (HLA) gene is one of the SNPs that is closely understood in a broad sense. In immune-mediated diseases in particular, there have been reports of SNPs that were associated with autoimmune diseases. It is known that the HLA and its costimulatory system form a necessary part of the immune response. People with certain HLAs are more likely to develop certain autoimmune diseases. For example, the HLA-DR3 allele was a shared SNP for Sjögren syndrome, diabetes mellitus type 1, and SLE [43,44]. An animal study showed that HLA-DR3 was associated with the autoimmune response induced by the anti-Smith (Sm) antibody in SLE patients [45]. Thus, the bio-function of the functional SNPs should also be verified through animal studies.

The present study has some merits. Previously, GO was mostly discussed with RA, and this study is the first research to show that SLE, RA, and GO share a genetic background. In addition, since the genotype frequency distribution of the control group was evaluated via HWE analysis before the genotype and haplotype analysis, this indicates that our findings are less prone to bias. The sample size of GO was a limitation, though FDR was used to correct for multiple testing, which could have solved the problem that the sample size of GO cases was small, which may have given false-negative outcomes. In addition, we also used a dual-luciferase reporter assay to verify the bio-functional effect of common SNPs on transcriptional activity. However, some limitations of the study should be acknowledged. This study only included the Taiwanese population. Thus, based on ethnic differences, the findings may only apply to the Taiwanese population. Additionally, because the materials of the reporter assay used in the promoter–reporter cannot be shared with the 3’UTR-reporter, and the 3’UTR-reporter assay needs to consider the influence of microRNA [46], only the promoter region was discussed in this study.

## 5. Conclusions

We found that there were six SNPs of genes that are involved in regulating T-cell activation that were common in SLE, RA, and GO. Furthermore, the bio-functional effect of the promoter SNPs on the transcriptional activity of the *CTLA4* gene was verified by dual-luciferase reporter assay. This indicated that these SNPs had a functional effect on the pathogenesis of autoimmune disease rather than just an association. Additionally, T-cell activation can be considered as an upstream pathway of adaptive immunity. Therefore, it can be inferred from this result that these SNP mutations involved in the upstream pathway of adaptive immunity may be related to the regulation of immune response, especially since these SNPs were also associated with other immune-related diseases or cancers.

## Figures and Tables

**Figure 1 biomedicines-11-02426-f001:**
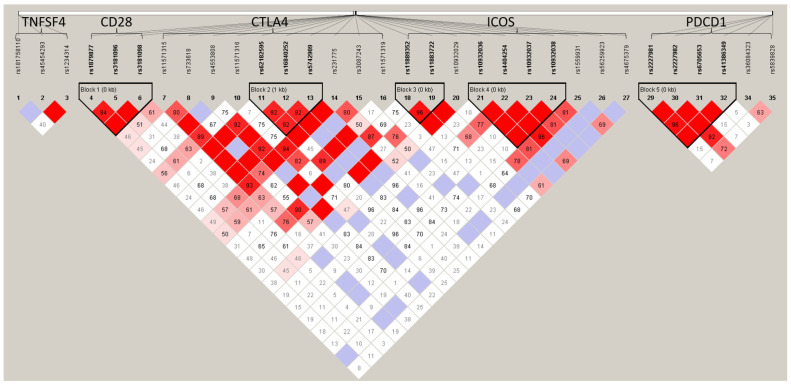
The linkage disequilibrium (LD) plot of the target genes for GO cases and controls. The color gradually changes from white to red, indicating that LD is becoming stronger, and purple indicates that there is no LD.

**Figure 2 biomedicines-11-02426-f002:**
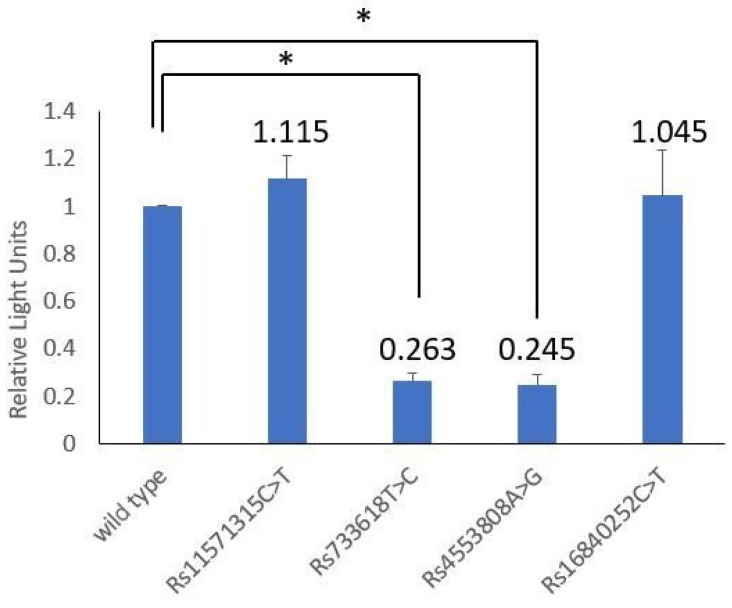
Comparing the transcriptional activity levels of the reporter contractions with rs11571315 C > T, rs733618 T > C, rs4553808 A > G, rs16840252 C > T and wild type. “*” indicates statistical significance (*p* < 0.05).

**Table 1 biomedicines-11-02426-t001:** The pairs of primers used for promoter–reporter construction with a single SNP variation.

Primer	Sequence	Position	Size
KpnI–CTLA4F	5′-ACAT GGTACC CTTGCTGCTAAGAGCATC-3′	203865939- 203867940	2001 bp
EcoRV–CTLA4R	5′-AGTA GATATCGGGCTTTATGGGAGCGGT-3′		
Rs11571315TF	5′-GCT CCT CTA CAT AAT ACT TCA A**T**T CCA GCA TTG-3′	203866178	
Rs11571315TR	5′-CAA TGC TGG A**A**T TGA AGT ATT ATG TAG AGG AGC-3′		
Rs733618CF	5′-TCA TGG GTT TAG CTG **C**CT GTC CCT GCC ACT-3′	203866221	
Rs733618CR	5′-AGT GGC AGG GAC AG**G** CAG CTA AAC CCA TGA-3′		
Rs4553808GF	5′-CAC TTT TT**G** AAA AAC CTC TGT TGC CCA GTC TGG C-3′	203866282	
Rs4553808GR	5′-GCC AGA CTG GGC AAC AGA GGT TTT T**C**A AAA AGT G-3′		
Rs16840252TF	5′-AAT GGG AAA CCA TGG A**T**G GAC TGG AGT AGG CA-3′	203866796	
Rs16840252TR	5′-TGC CTA CTC CAG TCC **A**TC CAT GGT TTC CCA TT-3′		

NCBI position was according to GRCh38.p13. The bold and underlined mutagenesis primer sequences were referred to as the position of site-directed mutagenesis.

**Table 2 biomedicines-11-02426-t002:** The HWE analysis in the control group and the allele frequencies in cases and controls.

SNP	Position	Allele	Minor Allele Frequency		HWE *p*-Value	Odds Ratio	*p^a^* Value
			Patient	Control		(95% CI)	
**CTLA4**							
rs11571315	203866178	**C**/T	0.163	0.4	0.925	0.291 (0.138–0.612)	0.001 *
rs733618	203866221	T/**C**	0.412	0.65	0.998	0.378 (0.199–0.717)	0.003 *
rs4553808	203866282	A/**G**	0.038	0.138	0.602	0.244 (0.065–0.912)	0.025 *
rs11571316	203866366	**A**/G	0.113	0.263	0.980	0.356 (0.152–0.836)	0.015 *
rs62182595	203866465	**A**/G	0.038	0.138	0.949	0.244 (0.065–0.912)	0.025 *
rs16840252	203866796	C/**T**	0.038	0.138	0.602	0.244 (0.065–0.912)	0.025 *
rs5742909	203867624	C/**T**	0.05	0.125	0.665	0.368 (0.111–1.228)	0.093
rs231775	203867991	**A**/G	0.213	0.355	0.990	0.530 (0.261–1.076)	0.077
rs3087243	203874196	G/**A**	0.163	0.263	0.980	0.545 (0.251–1.183)	0.122
rs11571319	203874215	G/**A**	0.075	0.313	0.102	0.178 (0.069–0.464)	<0.001 *
**CD28**							
rs1879877	203705277	**G**/T	0.321	0.449	0.759	0.580 (0.302–1.112)	0.100
rs3181096	203705369	C/**T**	0.175	0.269	0.036 *	0.576 (0.268–1.235)	0.154
rs3181097	203705416	**G**/A	0.325	0.615	0.988	0.301 (0.157–0.578)	<0.001 *
rs3181098	203705655	G/**A**	0.163	0.244	0.066	0.563 (0.257–1.230)	0.147
**PDCD1**							
rs5839828	241859601	**G**/GG	0.419	0.311	0.948	1.599 (0.814–3.140)	0.172
rs36084323	241859444	**C**/T	0.541	0.365	0.999	2.048 (1.061–3.955)	0.032 *
rs41386349	241851697	G/**A**	0.175	0.200	0.384	0.848 (0.383–1.880)	0.685
rs6705653	241851407	**T**/C	0.25	0.218	0.559	1.196 (0.572–2.503)	0.634
rs2227982	241851281	**G**/A	0.513	0.397	0.994	1.594 (0.848–2.995)	0.147
rs2227981	241851121	**A**/G	0.25	0.218	0.559	1.196 (0.572–2.503)	0.634
rs10204525	241850169	**C**/T	0.359	0.188	0.827	2.427 (1.172–5.023)	0.015 *
**ICOS**							
rs11571305	203935403	G/**A**	0.359	0.313	0.016 *	1.232 (0.636–2.387)	0.536
rs11889352	203935948	**T**/A	0.397	0.225	0.679	2.272 (1.135–4.546)	0.019 *
rs11883722	203936122	G/**A**	0.449	0.375	0.208	1.357 (0.718–2.561)	0.346
rs10932029	203937045	T/**C**	0.138	0.118	0.074	1.187 (0.462–3.047)	0.722
rs10932035	203959929	G/**A**	0.329	0.477	0.036 *	0.537 (0.251–1.149)	0.107
rs10932036	203960458	A/**T**	0.077	0.041	0.968	1.972 (0.475–8.193)	0.496
rs4404254	203960563	T/**C**	0.256	0.263	0.954	1.138 (0.558–2.322)	0.722
rs10932037	203960623	C/**T**	0.077	0.053	0.943	1.500 (0.406–5.541)	0.746
rs10932038	203960861	A/**G**	0.077	0.042	0.967	1.917 (0.461–7.967)	0.498
rs1559931	203961006	G/**A**	0.256	0.229	0.987	1.164 (0.547–2.475)	0.693
rs4675379	203961372	G/**C**	0.179	0.286	0.186	0.547 (0.226–1.325)	0.178
**TNFSF4**							
rs1234314	173208253	**C**/G	0.375	0.35	0.822	1.114 (0.585–2.124)	0.742
rs45454293	173208097	**A**/G	0.163	0.15	0.394	1.100 (0.468–2.583)	0.828

The position was obtained from Genome Assembly GRCh38.p13. HWE: Hardy–Weinberg equilibrium; 95% CI: 95% confidence interval; *p^a^* values of allele frequency were counted from the chi-square test or Fisher’s exact test. In the column of “Allele”, the bold refers to the minor allele, and the minor allele refers to the allele with lower frequency in the population containing cases and controls. “*”expresses *p* < 0.05.

**Table 3 biomedicines-11-02426-t003:** The significant SNPs associated with GO.

SNP	Genotype/Allele	Patient	Control	OR (95% CI)	*p*-Value	Q-Value
		*N* = 40	*N* = 40			
**CTLA4**						
rs11571315	CC vs. CT vs. TT				0.006 *	0.0720
	TT	28	15	Ref.	1.000	
	CT	11	18	0.327 (0.123–0.870)	0.023	0.1712
	CC	1	7	0.077 (0.009–0.682)	0.015	0.1675
	TT vs. CT + CC			0.257 (0.101–0.652)	0.004	0.0766
	TT + CT vs. CC			0.121 (0.014–1.034)	0.057	0.2634
rs733618	CC vs. CT vs. TT				0.011 *	0.0977
	CC	13	5	Ref.	1.000	
	CT	21	18	0.449 (0.134–1.502)	0.189	0.4715
	TT	6	17	0.136 (0.034–0.545)	0.003	0.0766
	CC vs. CT + TT			0.297 (0.094–0.934)	0.032	0.1787
	CC + CT vs. TT			0.239 (0.082–0.696)	0.007	0.1173
rs4553808	AA vs. AG vs. GG				0.019 *	0.0977
	AA	37	29	Ref.	1.000	
	AG	3	11	0.214 (0.055–0.838)	0.019 *	0.1675
	GG	0	0	NA	NA	
	AA vs. AG + GG			0.214 (0.055–0.838)	0.019 *	0.1675
	AA + AG vs. GG			NA	NA	
rs11571316	GG vs. AG vs. AA				0.056	0.1833
	GG	32	22	Ref.	1.000	
	AG	7	15	0.321 (0.112–0.916)	0.030 *	0.1787
	AA	1	3	0.229 (0.022–2.349)	0.305	0.6089
	GG vs. AG + AA			0.306 (0.113–0.826)	0.017 *	0.1675
	GG + AG vs. AA			0.316 (0.031–3.178)	0.615	0.8171
rs16840252	CC vs. CT vs. TT				0.019 *	0.0977
	CC	37	29	Ref.	1.000	
	CT	3	11	0.214 (0.055–0.838)	0.019 *	0.1675
	TT	0	0	NA	NA	
	CC vs. CT + TT			0.214 (0.055–0.838)	0.019 *	0.1675
	CC + CT vs. TT			NA	NA	
rs11571319	GG vs. AG vs. AA				<0.001 *	0.0178
	GG	34	16	Ref.	1.000	
	AG	6	23	0.123 (0.042–0.360)	<0.001 *	0.0302
	AA	0	1	NA	0.333	0.6375
	GG vs. AG + AA			0.118 (0.040–0.344)	<0.001 *	0.0302
	GG + AG vs. AA			NA	1.000	1.0000
**CD28**						
rs3181097	GG vs. AG vs. AA				<0.001 *	0.0178
	AA	15	6	Ref.	1.000	
	AG	24	18	0.533 (0.173–1.646)	0.271	0.5857
	GG	1	15	0.027 (0.003–0.249)	<0.001 *	0.0302
	GG vs. AG + AA			0.303 (0.103–0.892)	0.026 *	0.1742
	GG + AG vs. AA			0.041 (0.005–0.331)	<0.001 *	0.0302
rs3181098	GG vs. AG vs. AA				0.055	0.1833
	GG	27	25	Ref.	1.000	
	AG	13	9	1.337 (0.488–3.669)	0.572	0.8171
	AA	0	5	NA	0.053	0.2630
	GG vs. AG + AA			0.860 (0.339–2.180)	0.750	0.9105
	GG + AG vs. AA			NA	0.026 *	0.1742
**PDCD1**						
rs36084323	TT vs. CT vs. CC				0.110	0.2565
	TT	8	15	Ref.	1.000	
	CT	18	17	1.985 (0.671–5.871)	0.212	
	CC	11	5	4.125 (1.057–16.097)	0.037 *	
	TT + CT vs. CC			2.472 (0.890–6.864)	0.079	
	TT vs. CT + CC			2.708 (0.835–8.785)	0.090	
rs10204525	TT vs. CT vs. CC				0.075	0.2132
	TT	17	27	Ref.	1.000	
	CT	16	11	2.310 (0.868–6.146)	0.091	0.5073
	CC	6	2	4.765 (0.860–26.383)	0.118	0.1983
	TT + CT vs. CC			2.688 (1.076–6.715)	0.032 *	0.2786
	TT vs. CT + CC			3.455 (0.652–18.294)	0.154	0.2974
**ICOS**						
rs11889352	AA vs. AT vs. TT				0.045 *	0.1800
	AA	15	23	Ref.	1.000	
	AT	17	16	1.629 (0.635–4.183)	0.309	0.6089
	TT	7	1	10.733 (1.197–96.283)	0.020 *	0.1675
	AA vs. AT + TT			2.165 (0.881–5.322)	0.090	0.2974
	AA + AT vs. TT			8.531 (0.997–73.006)	0.029 *	0.1787
rs4675379	GG vs. CG vs. CC				0.042 *	0.1800
	GG	27	9	Ref.	1.000	
	CG	10	12	0.278 (0.090–0.859)	0.023 *	0.1712
	CC	2	0	NA	1.000	1.0000
	GG vs. CG + CC			0.333 (0.111–1.001)	0.047 *	0.2422
	GG + CG vs. CC			NA	0.537	0.7995

95% CI: 95% confidence interval; NA: not applicable. “*” expresses *p* < 0.05 and Q < 0.1.

**Table 4 biomedicines-11-02426-t004:** The significant haplotypes associated with GO.

Haplotypes	Freq. Cases	Freq. Controls	OR	95% CI	*p*-Value
CTLA4					
A_rs62182595_T_rs16840252_C_rs5742909_	0.075	0.250	0.243	0.061–0.964	0.034
A_rs62182595_T_rs16840252_T_rs5742909_	0.075	0.250	0.243	0.061–0.964	0.034
A_rs62182595_C_rs16840252_C_rs5742909_	0.075	0.250	0.243	0.061–0.964	0.034
A_rs62182595_C_rs16840252_T_rs5742909_	0.075	0.250	0.243	0.061–0.964	0.034
G_rs62182595_T_rs16840252_C_rs5742909_	0.075	0.250	0.243	0.061–0.964	0.034
ICOS					
A_rs11889352_C_11883722_	0.750	0.925	0.243	0.061–0.964	0.034

Freq.: frequency; OR: odds ratio; CI: confidence interval.

**Table 5 biomedicines-11-02426-t005:** Common SNPs in SLE, RA, and GO.

	GO Case	Control	SLE Case	Control	RA Case	Control
rs11571315						
TT	28	15	53	33	69	47
CT	11	18	15	31	33	41
CC	1	7	3	11	17	12
rs733618						
CC	13	5	33	15	36	18
CT	21	18	18	34	44	46
TT	6	17	21	26	41	36
rs4553808						
AA	37	29	71	55	103	77
AG	3	11	1	20	16	23
GG	0	0	0	0	4	0
rs16840252						
CC	37	29	69	53	105	75
CT	3	11	1	22	14	25
TT	0	0	1	0	4	0
rs11571319						
GG	34	16	58	40	91	61
AG	6	23	2	28	18	38
AA	0	1	8	7	15	1
rs36084323						
TT	8	15	19	33	32	40
CT	18	17	34	31	62	43
CC	11	5	18	7	30	13

**Table 6 biomedicines-11-02426-t006:** Analysis of the transcriptional activity of each common SNP variation in CTLA4 through dual-luciferase reporter assay.

Reporter Contractions	RLU						Mean	SD	*p*
CTLA4 wild type	1.00	1.00	1.00	1.00	1.00	1.00	1.00	0.00	Ref.
rs11571315 C > T	0.947	1.174	1.160	1.181	1.114	-	1.115	0.098	0.328
rs733618 T > C	0.218	0.293	0.234	0.291	0.256	0.286	0.263	0.032	<0.001 *
rs4553808 A > G	0.250	0.212	0.206	0.300	0.205	0.297	0.245	0.045	<0.001 *
rs16840252 C > T	0.819	1.093	0.799	1.188	1.141	1.232	1.045	0.189	0.929

SD: standard deviation. *: *p* < 0.05. Ref: reference.

## Data Availability

The datasets used and/or analyzed during the current study are available from the corresponding author on reasonable request.

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
