# Peer review of "Finding the Common Single-Nucleotide Polymorphisms in Three Autoimmune Diseases and Exploring Their Bio-Function by Using a Reporter Assay"

_biomedicines, 2023, doi:10.3390/biomedicines11092426_

Round 1
Reviewer 1 Report
This paper is about the detection of SNPs common to autoimmune diseases ~Grave's Disease and SLE, and RA, and that they affect the transcriptional activity of the CTLA4 gene, and two SNPs were observed to have a reduced activity of the gene.
The point of view is interesting and the experiment and analysis are well done. The results also led to the identification of some common SNPs.
HLA is one of the SNPs that is closely understood in a broad sense. In autoimmune diseases in particular, there have been reports of SNPs that are often specific to certain diseases (although there may be racial differences).
I would like to discuss the results of this study and HLA in the discussion.
Author Response
Thank you for the suggestion. It was known that HLA and costimulatory system are necessary part of immune response. And there have been reports of HLA SNPs that are shared among several autoimmune disease; for example, HLA-DR3 allele was a shared SNP for Sjögren syndrome, diabetes mellitus type 1, and SLE, and its function has been revealed, which helps us further understand the mechanism of disease development. Thus, the biofunction of the functional SNPs should also be verified through animal study. This discussion has been added in page 15, line 383-391.

Reviewer 2 Report
Dear Authors,
The article is very interesting.
Here are my point to point observations:
1. Tittle. I suggest using the entire name of “SNPs” as well.
2. Abstract. First line. What is “thyroid disease” since they are so many? Do you mean “autoimmune thyroid diseases”?
3. Abstract. Second sentence. The link with the other two rheumatologically conditions has been identified by more than one study.
4. Introduction. Line 38. Please correct “the study found” (since you cited 3 references) with “studies”.
5. Please explain abbreviations when first used such as “AITD”, etc.
6. Introduction. I suggest making it a clearer at Introduction – concerning the issue of the thyroid diseases. For example, “thyroid diseases are of various types, but some of the most important are autoimmune (namely autoimmune thyroid diseases). Among this class, there are 2 major types: Graves’s disease and Hashimoto’s autoimmune thyroiditis, two conditions situated at the end of the same spectrum. Eye involvement in GD, prior named Graves’s ophtalmopathy, is currently called “thyroid eye disease”. The specific antibody signature in GD is TRAb with stimulating effects, which is specific to this condition, also serving as prognostic marker. Studies have found a common link with other antibody – related non-endocrine conditions such as…”
7. Line 50. Many other more recent studies worth to be mentioned.
8. The last section of Introduction should be a clear sentence of the Objective (or Aim) of the present study
9. Methods. Firstly, we need to specify the type of study design.
10. Methods. The specific data concerning the studied population represents the first section at Results. First you apply the inclusion and exclusion criteria, and then you achieve the results after evaluation according to study’s assessments.
11. Discussion. For further studies, how would you expect the results in other populations?
12. Conclusion section should not start with “In summary” since it is the same.
Thank you
Minor editing is needed.
Author Response
- I suggest using the entire name of “SNPs” as well.
Response: Thanks for your suggestion. The “SNPs” in title has been edited to “single nucleotide polymorphisms”.
- First line. What is “thyroid disease” since they are so many? Do you mean “autoimmune thyroid diseases”?
Response: Yes. The “thyroid disease “ in the first line of abstract has been edited to “autoimmune thyroid diseases”. (page 1, line 13)
- Second sentence. The link with the other two rheumatologically conditions has been identified by more than one study.
Response: “A study” in the second sentence of abstract has been edited to “several studies”. (page 1, line 14)
- Line 38. Please correct “the study found” (since you cited 3 references) with “studies”.
Response: The “study” in line 39 has been edited to “studies”.
- Please explain abbreviations when first used such as “AITD”, etc.
Response: The full name of all abbreviations has been shown when it first occurrence. The full name of AITD has been shown on page 1, line 40.
- I suggest making it a clearer at Introduction – concerning the issue of the thyroid diseases. For example, “thyroid diseases are of various types, but some of the most important are autoimmune (namely autoimmune thyroid diseases). Among this class, there are 2 major types: Graves’s disease and Hashimoto’s autoimmune thyroiditis, two conditions situated at the end of the same spectrum. Eye involvement in GD, prior named Graves’s ophtalmopathy, is currently called “thyroid eye disease”. The specific antibody signature in GD is TRAb with stimulating effects, which is specific to this condition, also serving as prognostic marker. Studies have found a common link with other antibody – related non-endocrine conditions such as…”
Response: Thank you for the comment. The associated information has been supplemented in Introduction (page 2, line 58-60).
- Line 50. Many other more recent studies worth to be mentioned.
Response: Because it was the first prospective study of thyroid disorders in patients with SLE, it was mentioned.
- The last section of Introduction should be a clear sentence of the Objective (or Aim) of the present study
Response: The aim of this study has been added in the last section of Introduction. (page 2, line 78-81)
- Firstly, we need to specify the type of study design.
Response: Originally, it is a case-control study. We investigated the Graves’ ophthalmopathy susceptibility SNPs according to SNP analysis at first. Then, we observed that there were several common SNPs associated with SLE, RA, and GO. Thus, we tried to discuss the mechanism of these three autoimmune diseases caused by these common SNPs by using dual-luciferase reporter assay. Consequently, the type of the present study is case-control study plus functional study.
- The specific data concerning the studied population represents the first section at Results. First you apply the inclusion and exclusion criteria, and then you achieve the results after evaluation according to study’s assessments.
Response: Thank you for the suggestion. The specific data concerning the studied population has been removed to Result section. (page 4, line 161-163)
- For further studies, how would you expect the results in other populations?
Response: That's a good question. Because the disease susceptibility SNPs of other populations have not been verified for biological function, we cannot discuss them, but it can be said that the mechanism of other diseases related to the meaningful SNPs found in functional analysis may also due to their regulation of gene transcription activity. This opinion has been added in page 15, line 366-368.
- Conclusion section should not start with “In summary” since it is the same.
Response: Thanks for the suggestions. The “in summary” in Conclusion section has been deleted.

Reviewer 3 Report
In the present paper, the authors show that specific SNPs (rs11571315, rs733618, rs4553808, rs11571316, rs16840252, and rs11571319) of CTLA4, as well as rs3181098 of CD28, rs36084323, and rs10204525 of PDCD1, and rs11889352 and rs4675379 of ICOS, are significantly associated with GO, based on genotype and/or allele analysis (p<0.05). By consolidating the GO data with previously published findings on SLE and RA, they observed that rs11571315, rs733618, rs4553808, rs16840252, rs11571319, and rs36084323 were common across these three diseases. Furthermore, the authors tested the biological function of these SNPs through a dual-luciferase reporter assay, which revealed that rs733618 T>C and rs4553808 A>G significantly reduced the transcriptional activity (both p<0.001).
The study is interesting and the results worth of note. Limitations of the study are indicated in the Discussion section.
Minor revision
Author Response
Thank you for the compliment. And the English language of the article has been edited.
